



# Technical Note: Past and future warming – direct comparison on multi-century timescales

**Darrell S. Kaufman and Nicholas P. McKay**

School of Earth and Sustainability, Northern Arizona University, Flagstaff, AZ 86011, USA

**Correspondence:** Darrell S. Kaufman (darrell.kaufman@nau.edu)

**Abstract.** In 2013, the Intergovernmental Panel on Climate Change concluded that Northern Hemisphere temperatures had reached levels unprecedented in at least 1400 years. The 2021 report now sees global mean temperatures rising to levels unprecedented in over 100 000 years. This Technical Note briefly explains the reasons behind this major change. Namely, the new assessment reflects additional global warming that occurred between the two reports and improved paleotemperature reconstructions that extend further back in time. In addition to past and recent warming, the conclusion also considers multi-century future warming, which thereby enables a direct comparison with paleotemperature reconstructions on multi-century time scales.

## 1 Global paleotemperature assessment in AR6

The recent climate assessment report (AR6) by the Intergovernmental Panel on Climate Change (IPCC, 2021) concluded that global temperature[1] is reaching a level unprecedented in more than 100 000 years. That is much further back than was reported previously by the IPCC. The 2013 IPCC report (AR5) concluded that the Northern Hemisphere had warmed to levels that were unprecedented in at least 1400 years (Masson-Delmotte et al., 2013). With climate change presently on course to exceeding conditions uncharted by humans, the new report takes a longer-term view, with greater attention to paleoclimate reference periods further back in time (e.g., Box TS.2 in Arias et al., 2021; Cross-Chapter Box 2.1 in Gulev et al., 2021). AR6 also includes a better integration of paleoclimate across the report, including a direct comparison of paleotemperature and projected future temperature change (Fig. TS.1 in Arias et al., 2021; Cross-Chapter Box 2.1 Fig. 1 in Gulev et al., 2021). This Technical Note gathers evidence from several chapters of the new IPCC-AR6 Working Group I report and provides additional details for paleoclimate scientists to explain the basis for the updated assessment of recent and future global warming in a long-term context.

To assess the understanding of global temperature prior to industrialization, AR5, like its predecessors, relied heavily on annually resolved temperature records. The majority of these are tree-ring chronologies from the terrestrial Northern Hemisphere characterized by precise estimates of past climate variability with annual resolution. AR5 (Masson-Delmotte et al., 2013) focused on whether temperatures had yet exceeded those of medieval times, a period of relative warmth that is well known from some regions (PAGES 2k Consortium, 2013). Less attention was given to the Earth's temperature history prior to the last 2000 years because few tree-ring records are that long. Furthermore, when AR5 was drafted, global decadal average temperature was 0.8 °C higher than the 1850–1900 preindustrial reference period. In AR6, the last decade is estimated to have reached 1.1 °C warmer than this reference period. Of the 0.3 °C increase, 0.19 °C is the actual additional warming since the 2013 report; the other 0.1 °C is due to better methods and new data (Cross-Chapter Box 2.3 in Gulev et al., 2021). That is substantially higher, prompting us to look further back beyond the year-by-year reconstruction that is available over the past 2 millennia.

---

[1]The term "global temperature" is used to refer to both mean annual surface temperature and mean surface air temperature. AR6 determined that the two metrics differ by less than 10 % (Cross-Chapter Box 2.3 in Gulev et al., 2021).

Since 2013, there has been a major effort to assemble information from other archives of paleoclimate information prior to 2000 years ago. The most common sources are sediments that accumulate in lakes and in the ocean containing a variety of biogeochemical indicators of past temperature extending back through the Holocene and further (e.g., Kaufman et al., 2020a). Since AR5, the global coverage of these proxy records has increased, and their temporal resolution and age control has also improved. In addition, major new global data compilations have benefited from internationally coordinated campaigns and from better practices for data reuse (Williams et al., 2018). The cyberinfrastructure that supports the archival, distribution, interoperability, and reusability of these data has improved substantially. Notably, all of the site-level data and code used to reconstruct global temperature over the Holocene are now publicly available (see the "Code and data availability" section).

## 2   Temperature comparison on multi-century timescales

Sediment-based proxy records have a fundamental limitation, however; most are inherently smoothed so that the full extent of their fluctuations is attenuated relative to the decadal-scale climate fluctuations that actually occurred. The smoothing is caused by the physical mixing of sediment by burrowing organisms and by currents at the bottom of lakes and the ocean. It also arises because the age control for the sedimentary sequences has greater uncertainty than annually resolved records. When averaged together, finer-scale fluctuations are canceled out on average. As such, the Holocene global temperature reconstructions attest to temperature changes averaged over multiple centuries, which is a minimum estimate of changes that occurred on a decadal timescale.

Smoothing of the paleoclimate record makes it difficult to compare the long-term past temperature against the rapid changes of recent decades. However, these recent changes herald a long-term period of global warming that is unlikely to be reversed anytime soon, even under optimistically low-emissions scenarios. By looking both backward and forward, we can make informed estimates of long-term average temperature, thereby enabling a like-to-like, multi-century-scale comparison between the paleoclimate data and recent *plus* upcoming warming.

The upcoming global warming will be driven by continued emissions of greenhouse gases and will be exacerbated regionally as slow-moving components of the climate system continue to react to the greenhouse gases already in the atmosphere for centuries to come (Fox-Kemper et al., 2021). In AR6, climate model projections out to 2300 use new emissions scenario extensions (Meinshausen et al., 2020) and an updated emissions-driven emulator, which is calibrated to the extensively modeled 21st century climate (Lee et al., 2021).

These projections show that, even under a low-emissions scenario,[2] global temperature of at least 1 °C warmer than the late 1800s is nearly certain to persist for centuries (Fig. 1). Even for the low-emissions scenario (SSP1-2.6), the average temperature over the 400-year-long period from 1900 to 2300, which is a temporal resolution on par with proxy records, is estimated at 1.2 °C [0.9, 1.7 °C] warmer than 1850–1900 (mean [5 %, 95 % range]) (Lee et al., 2021). Assuming $CO_2$ emissions reach net zero, climate models indicate that global warming might still further increase or decrease during subsequent centuries (MacDougall et al., 2020). In contrast, under a high-emissions scenario (SSP3-7.0), the 400-year average temperature is estimated at 4.1 °C [2.1, 4.7 °C] relative to 1850–1900.

This multi-century, approximately 1.2 to 4.1 °C level of global warming can be reasonably compared with global temperature over a similar time horizon: that is, the warmest multi-century interval of the Holocene (Fig. 1). AR6 determined, with medium confidence, that peak Holocene global temperature was between 0.2 and 1 °C warmer than the late 1800s (Gulev et al., 2021). This assessed temperature range was based on several published studies of proxy-based temperature reconstructions. These include a global compilation of quality-controlled marine and terrestrial multi-proxy records (Temp12k, Kaufman et al., 2020a), which was used to generate a multi-method ensemble global temperature reconstruction (Kaufman et al., 2020b). The median reconstruction shows that the warmest 200-year-long interval prior to industrialization took place around 6500 years ago, when global temperature is estimated to have been 0.7 °C [0.3, 1.8 °C] warmer than the 19th century. The Temp12k reconstruction is similar to the only other available Holocene global temperature reconstruction based on both terrestrial and marine proxy data (Marcott et al., 2013). That study used a much smaller dataset and different procedures to estimate maximum warmth of 0.8 ± 0.3 °C (2σ) relative to 1850–1900 at around 7 ka (adjusted by adding 0.3 °C to account for different reference periods). In contrast, AR5-generation reconstructions based on selected proxy types indicate that mid-Holocene land and ocean surface temperatures were indistinguishable from preindustrial climate (Harrison et al., 2015). This and other evidence for relatively low mid-Holocene global temperatures led to the AR6-assessed

---

[2]Based on information in Lee et al. (2021), in the low-emissions scenario (SSP1-2.6), emissions peak in this decade and then decrease to net negative emissions by 2080. In the intermediate scenario (SSP2-4.5), emissions remain at about the current levels until the middle of the century, then decrease. In the high-emissions scenario (SSP3-7.0), emissions approximately double between 2015 and 2100. Following 2100, all scenarios reduce emissions such that $CO_2$ emissions from land use are reduced to zero by 2150, any net negative fossil $CO_2$ emissions are reduced to zero by 2200, and positive fossil $CO_2$ emissions are reduced to zero by 2250. Non-$CO_2$ fossil fuel emissions are also reduced to zero by 2250, while land-use-related non-$CO_2$ emissions are held constant at 2100 levels.

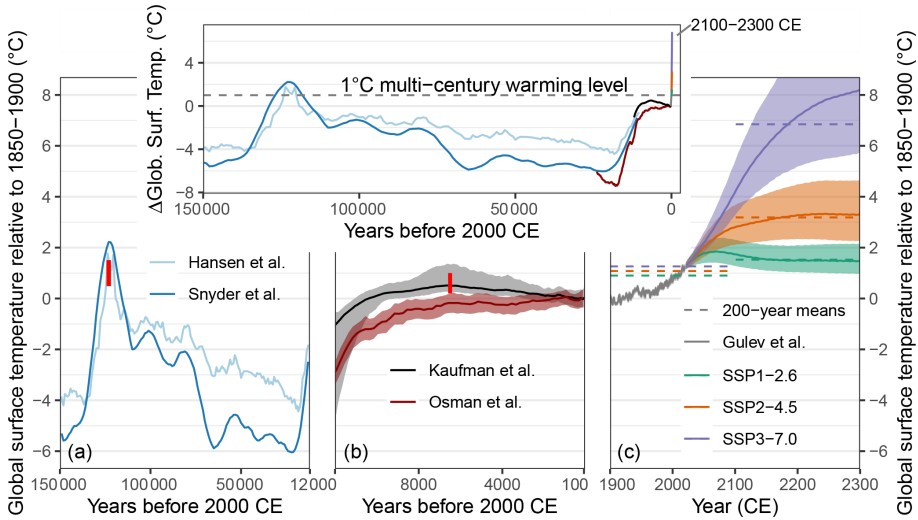

**Figure 1.** Global surface temperature over 150 000 years relative to 1850–1900. Three timescales are shown according to their published resolutions: **(a)** 150 000 to 12 000 years ago based on marine oxygen isotopes in foraminifera (Hansen et al., 2013; Snyder, 2016), **(b)** 12 000 years ago to 1900 CE based on the Temp12k multi-method, multi-proxy ensemble reconstruction (Kaufman et al., 2020b) along with a post-AR6 reconstruction that blends marine proxy evidence with a climate model simulation (Osman et al., 2021a), and **(c)** 1900 to 2300 based on instrumental observations (gray line) (Gulev et al., 2021) and projections using three emissions scenarios (low, intermediate, and high) as assessed in AR6 (Lee et al., 2021). Horizontal dashed lines are 200-year averages (1900–2100, 2100–2300). Lines and shading in **(b)** and **(c)** show the ensemble mean and the 5 %–95 % range. Red bars in **(a)** and **(b)** are the IPCC-assessed (medium confidence; Gulev et al., 2021) temperature ranges from proxy evidence for the warmest intervals of the last interglacial (around 125 000 years ago, 0.5–1.5 °C) and Holocene (around 6500 years ago, 0.2–1 °C). These values are based on multiple studies, including the time series displayed here. The inset shows all three panels on the same timescale with 200-year resolution so that the past, present, and future are directly comparable. See the "Code and data availability" section for details on data sources and adjustments used to align temperature changes relative to the 1850–1900 reference period.

value that favors the lower estimates within the Temp12k ensemble reconstruction (Fig. 1b).

Climate models generally simulate Holocene global temperature that correlates with greenhouse gas concentrations. More specifically, global temperature estimated from 16 climate models programmed to simulate climate for the mid-Holocene, 6000 years ago, averages around $-0.3 \pm 0.1$ °C ($\pm 1\sigma$) colder than preindustrial control runs (Brierley et al., 2020). A recent global temperature reconstruction based on data assimilation, which was published after AR6, blends climate model simulations with sea-surface temperature proxies and shows essentially no multi-millennial change in global temperature following 8000 years ago and prior to industrialization (Osman et al., 2021a). While the discrepancy between the observational and model results is the subject of ongoing research, the lower temperatures simulated by models are consistent with the conclusion that a global warming level of at least 1 °C is unprecedented during the preindustrial Holocene.

We need to look much further back for a time when temperature might have exceeded the 1 °C global warming level (Fig. 1). There is no evidence that global temperature higher than Holocene occurred during the last major ice age (MISs 4–2; MIS: marine isotope stage). For the last interglacial (MIS 5) around 125 000 years ago, AR6 (Gulev et al., 2021) has medium confidence that global temperature averaged over multiple centuries was between 0.5 and 1.5 °C higher than the late 1800s, which overlaps with the 1 °C warming level. This led to the AR6 statement that the last decade was more likely than not warmer than any multi-century period after the last interglacial (Gulev et al., 2021). The statement was simplified in Fig. SPM.1 of the Summary for Policy Makers (IPCC, 2021), which stated that global temperature during the mid-Holocene was the warmest in at least the last 100 000 years and that the last interglacial, around 125 000 years ago, is the next most recent candidate for a period of higher temperature. No attempt was made in AR6 to assess temperatures of other warm substages of MIS 5.

## 3 Conclusion

Global warming has reached 1 °C relative to the late 1800s and, in the absence of a strong reduction in greenhouse gas emissions, is on track to remain at least as warm and possibly much warmer for multiple centuries. The duration of this ongoing and upcoming global warming is on par with that of the resolution of Holocene paleotemperature reconstructions. Human-caused global warming is now exceeding the warmest multi-century period of the Holocene and thereby the envelope of temperatures under which agricultur-

ally based society has flourished (Steffen et al., 2018). Without rapid and sustained reductions in greenhouse gas emissions, the average temperature of coming centuries will exceed 1.5 °C above preindustrial temperature and will therefore be warmer than the peak of the last interglacial around 125 000 years ago.

**Code and data availability.** The time series data plotted in Fig. 1 are available through publicly accessible sources as specified below. These and other Cenozoic paleotemperature reconstructions are compared in Gulev et al. (2021, Cross-Chapter Box 2.1). Temperatures were adjusted to a common baseline (1850–1900, as specified below) with a uniform representation of uncertainties (5 %–95 % ensemble range), and the values were compiled into a single table for convenience of future use (see the Supplement). The data and code used to draft the figure are available at https://doi.org/10.5281/zenodo.5842208 TS1 (McKay, 2022).

- Dataset: Late Quaternary global temperature according to equations by Hansen et al. (2013) applied to benthic marine oxygen isotope stack of Zachos et al. (2008).
  Available at http://www.columbia.edu/~mhs119/Sensitivity+SL+CO2/Table.txt (last access: 21 March 2022).
  Modified: subtracted 14.15 °C to adjust temperature relative to 1961–1990 and added 0.36 °C to adjust to 1850–1900 reference period based on the AR6-assessed four-dataset mean (Trewin, 2022; GMST-component_data_sets.csv).

- Dataset: Late Quaternary multi-proxy sea surface temperature stack converted to global temperature by Snyder (2016).
  Available at https://static-content.springer.com/esm/art%3A10.1038%2Fnature19798/MediaObjects/41586_2016_BFnature19798_MOESM258_ESM.xlsx (last access: 21 March 2022).
  Modified: added 0.23 °C to adjust late Holocene temperature to the 1850–1900 reference period based on the reconstruction of Kaufman et al. (2020b).

- Dataset: Holocene multi-method ensemble global temperature of Kaufman et al. (2020b) based on multi-proxy marine and terrestrial paleotemperature data (Temp12k; Kaufman et al., 2020a) using code of Rouston TS2 et al. (2020, https://doi.org/10.5281/zenodo.3888590 TS3).
  Available at https://www.ncei.noaa.gov/access/paleo-search/study/29712 (Kaufman et al., 2020c; temp12k_allmethods_percentiles).
  Modified: subtracted 0.03 °C to adjust 19th century mean temperature to 1850–1900 reference period based on PAGES 2k Consortium (2019) 10-year smoothed multi-method reconstruction (Gilbert et al., 2021 TS4; SPM1_1-2000.csv).

- Dataset: Last Glacial Maximum reanalysis (LGMR) of Osman et al. (2021a) based on marine paleotemperature data (Osman et al., 2021b; proxyDatabase.nc) assimilated using climate model (iCESM) priors and code from https://github.com/JonKing93/DASH TS5 v.3.6.1.
  Available at https://www.ncei.noaa.gov/access/paleo-search/study/33112 (Osman et al., 2021b; LGMR_GMST_ens.nc).
  Modified: subtracted 13.49 °C (median of the most recent bin) to adjust temperature relative to 1750–1950 and added 0.03 °C to adjust to 1850–1900 reference period based on PAGES 2k Consortium (2019) 10-year smoothed multi-method reconstruction (Gilbert et al., 2021 TS6; SPM1_1-2000.csv).

- Dataset: 1850–2020 global temperature of Gulev et al. (2021) based on mean of four instrumental datasets (HadCRUT, NOAA, Berkeley Earth, Kadow) assessed by IPCC-AR6-WGI and shown in Fig. 2.11c.
  Available at https://doi.org/10.5281/zenodo.6321535 (Trewin, 2022; GMST-component_data_sets.csv).
  Modified: none (1850–1900 reference period).

- Dataset: Global temperature projections to 2300 of Lee et al. (2021) based on the MAGICC (v.7.5.0) emulator (Meinshausen et al., 2020) calibrated against the IPCC-AR6assessed temperature to 2100 and shown in Fig. 4.40a TS7.
  Available at https://doi.org/10.5281/zenodo.6386979 (Nicholls et al., 2022; files with titles containing "fig-4-40" and respective SSP identifiers).
  Modified: none (1850–1900 reference period).

**Supplement.** The supplement related to this article is available online at: https://doi.org/10.5194/cp-18-1-2022-supplement. TS8

TS9

**Author contributions.** DSK wrote the paper with input from NPM. NPM drafted the figure.

**Competing interests.** The contact author has declared that neither they nor their co-author has any competing interests.

**Disclaimer.** Publisher's note: Copernicus Publications remains neutral with regard to jurisdictional claims in published maps and institutional affiliations.

**Acknowledgements.** We thank Dan Lunt and an anonymous reviewer for their helpful comments.

**Financial support.** This research has been supported by the National Science Foundation (grant no. 1929460).

**Review statement.** This paper was edited by Nerilie Abram and reviewed by Dan Lunt and one anonymous referee.

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

TS2    Please confirm name.

TS3    According to our standards the link to the code needs to be added in this section.

TS4    This reference is not in the reference list. Please add it.

TS5    Please clarify whether the code is your own. If yes, please provide a DOI in addition to your GitHub URL since our reference standard includes DOIs rather than URLs. If you have not yet created a DOI for your code, please issue a Zenodo DOI (https://help.github.com/en/github/creating-cloning-and-archiving-repositories/referencing-and-citing-content). If the code is not your own, please inform us accordingly. In any case, please ensure that you include a reference list entry corresponding to the code including creators, title, and date of last access.

TS6    This reference is not in the reference list. Please add it.

TS7    Please clarify in which publication this figure can be found.

TS8    Please note that the supplement file you sent had not only the "Read me" tab changed but was completely different. Please check if you sent the correct file. If so, please note that we cannot simply replace the supplement file at this stage; changes like this require the editor's approval. Therefore, please provide a detailed explanation for the changes in the new file that can be forwarded to the editor.

TS9    I am sorry, but I did not find a reference to the supplement in your explanation for the editor. Please check.

TS10    Please note that the new DOI leads to a Zenodo page with the following citation: "McKay, N. and Kaufman, D.: nickmckay/past-and-future-warming-comparison-figure code repository. (page proofs), Zenodo [code], https://doi.org/10.5281/zenodo.6399351, 2022.". This citation uses a completely different DOI. Please check and update reference list entry.