# Peer review of "Technical Note: Past and future warming – Direct comparison on multi-century timescales"

_Climate of the Past, 2021_

## Referee Comment (RC2)

This paper aims to provide some peer-reviewed support for one of the high-level IPCC assessment statements, namely that current temperatures are higher than any previous temperatures of the last (at least) 100,000 years.

I originally reviewed this paper for another journal, and many of my previous review comments have been addressed in this version, but here are some new/outstanding comments:

[I should also comment that I didn't read the other review prior to completing this one, so my review is independent]

My two main comments are that:

(1) it is not always clear in the paper when the present (recent) warming is being compared with the past, and when future warming is being compared with the past. This should be delineated more clearly. For example, in the abstract the first four lines are all about paleo and recent warming, whereas the final sentence suddenly brings in future warming, which is not mentioned previously. Also line 22 could mention future as well as recent warming. The concept of "upcoming" warming is introduced later in the paper, but I am still unsure whether the actual assessment statement relates to warming that has happened already, or warming that may happen in the future.

(2) I feel that in a "Technical Note" such as this, it would be good to have a bit more technical detail. For example, exactly what are the new papers that have come out since AR5 that allow us to go further back in time? It would be great of these were listed in e.g. a table, and the time period at which they give a paleo warming of 1.1 degrees given. This would allow us to clearly understand the new assessment in IPCC. In other words, I would have hoped that by the end of reading the Technical Note I would fully understand why IPCC assessed a date of 100,000 years, rather than 110,000 years, or 90,000 years. Indeed, looking t Figure 1 I would have expected an older date than 100,00, more like ~120,000 years.

In addition:

- it might be good to add a sentence or so stating that the new structure of AR6, with paleo integrated throughout the report, was one of the contributing factors that facilitated this new assessment of future warming with paleo time periods (if the authors believe this to be the case).
- The abstract should be more quantitative, in particular including the 1.1°C of recent warming, the 1.5°C maximum warming of the last interglacial, and the 1°C maximum warming of the Holocene.
- Page 2, line 53-57, "This includes committed climate change, which arises because slow-moving components of the climate system will continue to react to the greenhouse gasses already in the atmosphere for centuries to come.". Yes, but bear in mind that after zero emissions is achieved, $CO_2$ starts to decrease, and this largely cancels out the effect you mention, so that committed "warming" is actually quite small. See MacDougall et al, 2020.

MacDougall, A. H., Frölicher, T. L., Jones, C. D., Rogelj, J., Matthews, H. D., Zickfeld, K., Arora, V. K., Barrett, N. J., Brovkin, V., Burger, F. A., Eby, M., Eliseev, A. V., Hajima, T., Holden, P. B., Jeltsch-Thömmes, A., Koven, C., Mengis, N., Menviel, L., Michou, M., Mokhov, I. I., Oka, A., Schwinger, J., Séférian, R., Shaffer, G., Sokolov, A., Tachiiri, K., Tjiputra , J., Wiltshire, A., and Ziehn, T.: Is there warming in the pipeline? A multi-model analysis of the Zero Emissions Commitment from CO2, Biogeosciences, 17, 2987–3016, https://doi.org/10.5194/bg-17-2987-2020, 2020.

---

## Author Response (AR1)

Response to reviewers

Anonymous Referee 1
We appreciate the Reviewer's positive words about the summary and their comment about committed warming. In our revised manuscript, we will remove the statement about committed warming and replace it with, "Assuming $CO_2$ emissions reach net zero, climate models indicate that global warming might still further increase or decrease during subsequent centuries (MacDougall et al., 2020)". As suggested by the Reviewer, we will add the new temperature reconstruction by Osman et al. (2021) to the text and to the figure, and will make the other minor changes in wording suggested by the Reviewer.

Referee 2 (Lunt)
We appreciate the Reviewer's insightful suggestions. Regarding the two main comments:
(1) We will revise the wording in the Abstract and on line 22 to be more explicit about future warming. (2) We will add additional details about the specific individual papers that underlie the AR6 assessed temperature for the mid-Holocene. This will explain why the value is narrower than the 5-95% range of the ensemble reconstruction shown in Fig. 1B. We will also explain the rationale behind the assessed dates of 100,000 years and 125,000 years. Regarding the additional three points: (a) We will mention that paleoclimate topics are integrated across AR6. (b) We prefer to not specify the temperature values in the Abstract because each of values is associated with some caveats, which we hesitate to oversimplify. (c) We will remove the statement about committed warming as explained in our reply to Referee RC1.